# Lysicamine Reduces Protein Kinase B (AKT) Activation and Promotes Necrosis in Anaplastic Thyroid Cancer

**DOI:** 10.3390/ph16121687

**Published:** 2023-12-04

**Authors:** Mariana Teixeira Rodrigues, Ana Paula Picaro Michelli, Gustavo Felisola Caso, Paloma Ramos de Oliveira, Dorival Mendes Rodrigues-Junior, Mirian Galliote Morale, Joel Machado Júnior, Karina Ramalho Bortoluci, Rodrigo Esaki Tamura, Tamiris Reissa Cipriano da Silva, Cristiano Raminelli, Eric Chau, Biana Godin, Jamile Calil-Silveira, Ileana G. Sanchez Rubio

**Affiliations:** 1Thyroid Molecular Sciences Laboratory, Universidade Federal de São Paulo—UNIFESP, São Paulo 04021-001, Brazil; mariana.rodr@outlook.com (M.T.R.); anapaulamichelli@hotmail.com (A.P.P.M.); gustavocaso78@gmail.com (G.F.C.); paloma.oliveira24@unifesp.br (P.R.d.O.); jamile.calil@gmail.com (J.C.-S.); 2Structural and Functional Biology Post-Graduate Program, Universidade Federal de São Paulo—UNIFESP, São Paulo 04021-001, Brazil; 3Cancer Molecular Biology Laboratory, Universidade Federal de São Paulo—UNIFESP, São Paulo 04021-001, Brazil; morale@unifesp.br (M.G.M.); rodrigo.tamura@unifesp.br (R.E.T.); 4Department of Medical Biochemistry and Microbiology, Science for Life Laboratory, Biomedical Center, Uppsala University, 752 36 Uppsala, Sweden; dorivalmrjr@gmail.com; 5Biological Science Department, Universidade Federal de São Paulo—UNIFESP, Diadema 09920-000, Brazil; joel.jr@unifesp.br; 6Pharmacology Department, Escola Paulista de Medicina, Universidade Federal de São Paulo—UNIFESP, São Paulo 04021-001, Brazil; karina.bortolucci@unifesp.br; 7Biology–Chemistry Post-Graduate Program, Institute of Environmental, Chemical and Pharmaceutical Science, Universidade Federal de São Paulo—UNIFESP, Diadema 09920-000, Brazil; 8Department of Chemistry, Institute of Environmental, Chemical and Pharmaceutical Science, Universidade Federal de São Paulo—UNIFESP, Diadema 09920-000, Brazil; tamiris.cipriano@gmail.com (T.R.C.d.S.); raminelli@unifesp.br (C.R.); 9Department of Nanomedicine, Houston Methodist Research Institute, Houston, TX 77030, USA; echau@houstonmethodist.org (E.C.); bgodin@houstonmethodist.org (B.G.); 10Department of Obstetrics and Gynecology, Weill Cornell Medicine College, New York, NY 10065, USA; 11Health Board III, Universidade Nove de Julho, São Paulo 01525-000, Brazil

**Keywords:** thyroid cancer, anaplastic thyroid cancer, lysicamine, natural compounds, necrosis, akt phosphorylation

## Abstract

Anaplastic thyroid cancer (ATC) is an aggressive form of thyroid cancer (TC), accounting for 50% of total TC-related deaths. Although therapeutic approaches against TC have improved in recent years, the survival rate remains low, and severe adverse effects are commonly reported. However, unexplored alternatives based on natural compounds, such as lysicamine, an alkaloid found in plants with established cytotoxicity against breast and liver cancers, offer promise. Therefore, this study aimed to explore the antineoplastic effects of lysicamine in papillary TC (BCPAP) and ATC (HTH83 and KTC-2) cells. Lysicamine treatment reduced cell viability, motility, colony formation, and AKT activation while increasing the percentage of necrotic cells. The absence of caspase activity confirmed apoptosis-independent cell death. Necrostatin-1 (NEC-1)-mediated necrosome inhibition reduced lysicamine-induced necrosis in KTC-2, suggesting necroptosis induction via a reactive oxygen species (ROS)-independent mechanism. Additionally, in silico analysis predicted lysicamine target proteins, particularly those related to MAPK and TGF-β signaling. Our study demonstrated lysicamine’s potential as an antineoplastic compound in ATC cells with a proposed mechanism related to inhibiting AKT activation and inducing cell death.

## 1. Introduction

Thyroid cancer (TC) is the ninth most common cancer, and its incidence is continuously increasing worldwide [1,2]. This rise is attributed to overdiagnosis [3,4] as well as to various risk factors, including obesity, radiation exposure, and circulating levels of iodine and estrogen; the latter contributes to an incidence 3-fold higher in women than in men [5,6,7,8]. Papillary (PTC) and follicular (FTC) TC, referred to as differentiated thyroid cancer (DTC), correspond to most TC cases (80% and 10%, respectively). DTC is usually responsive to treatment and has a 5-year survival rate above 90%, yet 10% of DTC patients develop an aggressive and radioresistant form, leading to a worse prognosis of TC [9,10]. Anaplastic TC (ATC) is a rare and aggressive type of undifferentiated TC (1–2% of cases), among which almost half of the patients have metastasis at diagnosis and a low survival rate (0.5–3 years) [11,12]. Hence, aggressive DTC and ATC combined are responsible for more than 50% of TC-related deaths [13]. 

Currently, receptor tyrosine kinase inhibitors (TKI), such as sorafenib (antiangiogenic and BRAF inhibitor) and levantinib (antiangiogenic), are used in the therapy of radioresistant DTC [14]. In addition, two potential RET kinase inhibitors, selpercatinib and pralsetinib, are in the clinical trial phase for the treatment of DTCs with fusions in the RET gene [15]. In 2016, dabrafenib and trametinib, BRAF, and MEK inhibitors, respectively, were approved by the Food and Drugs Administration (FDA) for ATC harboring BRAF^V600E^ mutation. Although the combination of this regimen reduced deaths caused by ATC, patients still progress poorly, as shown in 48% of ATC patients developing metastases or locoregional progression [16]. As promising as the results in clinical studies are, many patients experience significant adverse effects [17]. As an example, despite being effective in restoring iodine uptake in tumor cells, a combination of selumetinib and dabrafenib caused significant hematological toxicities due to the accumulation of radioactive iodine (I^131^) [18]. Therefore, discovering new, safe, and more effective compounds for TC therapy is a major unmet medical need.

Given the substantial number of drugs derived from herbal medicine and natural products, it is reasonable to explore the potential of herbs used in Traditional Chinese Medicine (TCM) as sources for developing novel and promising treatments [19]. Aporphine alkaloids are bioactive compounds isolated from plants, such as *Liriodendron tulipifera*, *Artabotrys crassifolius*, and *Nelumbo nucifera gaertn*, or obtained by semi- or total synthesis with multiple pharmacological properties, including antioxidant, anti-inflammatory, and anti-cancer activities [20,21,22]. 

Among the aporphine alkaloids, lysicamine, an oxoaporphine alkaloid, has been reported to inhibit the growth of breast and hepatocellular carcinoma cells [23] but not prostate and gastric cancer cells [24]. Reduced cytotoxicity of lysicamine was observed in kidney Vero cells [25] and in human neuroblastoma cells [26]. Lysicamine also showed moderate ferric reducing power [24], anti-bacterial activity [23], and neuroprotective effect against amino-chrome-induced cytotoxicity [26]. No previous information regarding its effects on thyroid cancer cells has been reported. Therefore, our study aimed to evaluate the impact of lysicamine therapy on DTC and ATC cells.

## 2. Results

### 2.1. Lysicamine Induces Cell Toxicity in Monolayer Cell Culture

Lysicamine was synthesized via benzyne chemistry, as we previously described [27] (Figure 1a). Changes in cell viability were evaluated in PTC (BCPAP), ATC (HTH83 and KTC-2), and in immortalized non-tumoral thyroid follicular epithelial (Nthy-ORI) cell lines 72 h following lysicamine treatment. Increasing concentrations of lysicamine significantly reduced viability in all cell lines. KTC-2 was the most sensitive cell line (Figure 1b). The IC_50_ value was subsequently determined from a dose–response curve after 72 h treatment. Lysicamine IC_50_ values were 15.6 µM, 36.4 µM, 30.5 µM, and 30.9 µM for KTC-2, HTH83, BCPAP, and Nthy-ORI, respectively, with a selectivity index (SI) of 1.98, 0.85, and 1.01, respectively (Table 1). As a positive control, IC50 values of Cisplatin were 1.9 µM, 3.6 µM, and 2.2 µM for KTC-2, HTH83, and BCPAP, respectively.

Clonogenic assays showed that lysicamine applied at IC50 concentration to the cells in culture reduced the number of colonies formed in KTC-2, HTH83, and BCPAP cells approximately by 95%, 64%, and 91%, respectively (Figure 1c). Additionally, the wound-healing assay suggested lysicamine slightly reduced the migration capacity of KTC-2 cells by 10% after 12 h treatment (Figure 1d). Together, these results point toward the viability-reducing effect of lysicamine in all TC cell lines, with KTC2 being the most sensitive cell line.

### 2.2. Lysicamine Induces Cell Toxicity in the Tumor Spheroids

Since the tumors are not “flat” and have 3D architecture that imparts transport barriers to therapies, drugs usually are more effective in monolayer cell cultures than in 3D spheroid tumor models [28]. Thus, it was important to assess the effect of lysicamine in 3D TC cultures. KTC-2 and HTH83 tumor spheroids were assembled using a modified hanging drop technique and treated with lysicamine after formation. Treatment with the IC_50_ of lysicamine for 72 h had no effect on the spheroids’ viability in both cell lines. No effect was observed 7 and 14 days after exposure of the tumor spheroids to the compound at IC_50_. For this reason, we doubled the concentration to assess the effect of lysicamine on the cell viability at 10–14 days after the exposure. In this setting, 26.5% reduced viability was observed in KTC-2 14 days after treatment (*p* < 0.05, Figure 2a). In HTH83, 2xI IC_50_ reduced viability by 32.4% and 57.4% 10 and 14 days after the incubation with lysicamine, respectively (*p* < 0.05, *p* < 0.0001, Figure 2b). 

We also analyzed the TC spheroid area on days 0, 3, 7, and 10. In KTC-2, the spheroid area was 31.4% and 75.4% smaller on days 7 and 10, respectively, after 2× IC_50_ lysicamine treatment (*p* < 0.05, *p* < 0.0001, respectively, Figure 2c,d). Area reduction was also observed in HTH83 spheroids, as 62.0% on day 10 (*p* < 0.0001, Figure 2e,f).

### 2.3. Lysicamine Promoted Necroptotic Cell Death in a Reactive Oxygen Species (ROS)-Independent Manner

To acquire insights into lysicamine-induced toxicity in TC cell lines, cells were stained with Annexin V (AV) and 7-Aminoactinomycin D (7-AAD). Lysicamine administration for 48 h at the IC_50_ led to a reduction of 51.7%, 45.4%, and 37.6% of live cells (AV −/7-AAD−) in KTC-2, HTH83, and BCPAP, respectively (Figure 3a). Furthermore, an increase of 44.1%, 36.9%, and 28.8% of 7-AAD-stained cells (AV−/7-AAD+) was observed, suggesting that lysicamine treatment induces necrosis (Figure 3b). No effect on the population of AV-positive cells was observed (Appendix A). No significant increase in caspase 3/7 activity was measured in all TC (Figure 3c), confirming that apoptosis was not involved in the TC cell death following lysicamine treatment. 

To explore the potential anti- or pro-oxidant role of lysicamine and the involvement of reactive oxygen species (ROS) in cell death, we evaluated the ROS generation via the 2′,7′-Dichlorofluorescin diacetate (DCF-DA) reagent. A significant increase in ROS levels was observed in HTH83 cells in a dose-dependent manner, while no effect on ROS modulation was noted in KTC-2 and BCPAP cells treated with lysicamine (Figure 3d,e). To verify whether HTH83 necroptotic cell death mediated by lysicamine was dependent on ROS induction, a co-treatment with a ROS scavenger, N-acetylcysteine (NAC), was performed, but the percentage of necrotic cells (AV−/7-AAD+) was not altered (Figure 3f), indicating that ROS induction in HTH83 does not play a role in the necrosis mediated by lysicamine.

Finally, we assessed whether lysicamine could induce necroptosis, the regulated form of necrosis. KTC-2 cells were treated with lysicamine in the presence or absence of Nec-1, a necrosome inhibitor. A reduction of 11.4% in necrotic cells (AV−/7-AAD+) was observed (Figure 3e). Together, our data strengthened the evidence that lysicamine induces cell death in TC cells via necroptosis in a ROS-independent manner.

### 2.4. Lysicamine Modulates AKT Phosphorylation

Next, the question was raised regarding whether lysicamine could modulate the most commonly activated pathways in TC, specifically the MAPK and PI3K/AKT pathways [29]. Western blot data show that in KTC-2 cells, AKT phosphorylation (p-AKT) was reduced in a dose-dependent manner following 48 h treatment with IC_50_ and 1.5× IC_50_ lysicamine (Figure 4a). In HTH83 cells, only 1.5× IC_50_ lysicamine treatment reduced p-AKT (Figure 4b). There was no effect on ERK phosphorylation in both cell lines. These results indicate that lysicamine reduced PI3K/AKT pathway activation while did not modulate the MAPK pathway (Figure 4a,b).

### 2.5. In Silico Prediction of Pharmacological Activity and Interacting Proteins for Lysicamine

To explore and expand our understanding of the lysicamine mechanism of action, its biological activity was characterized based on the chemical structure (Figure 1a), and direct chemical–protein interactions were predicted using two independent in silico analyses. By applying the PASS tool [30], a spectrum of pharmacological activities was predicted, including CYP2E1 inducer, JAK2 expression inhibitor, Cytochrome P450 stimulant, and beta-tubulin antagonist (Figure 5a). Moreover, a total of 306 proteins were predicted to interact directly with lysicamine and among the top 10 proteins with higher confidence (>0.4) the MAPK family proteins (MAPK3, MAP2K7, and MAPK12) were highlighted, besides the Serine/threonine–protein kinase PLK3 and the Cyclin-dependent kinase 13 (CDK13) and 19 (CDK19) (Figure 5b, Appendix A). Additionally, 46 proteins interacting with lysicamine were predicted based on the SEA database [31] and Tubulin beta-1 chain (TUBB1), Potassium voltage-gated channel subfamily B member 1 (KCNB1), Cytochrome P450 1B1 (CYP1B1), and the Receptor-type tyrosine–protein phosphatase C (PTPRC) were among the top 10 proteins with high probability of such direct interaction (Figure 5c, Appendix A). Of note, seven proteins were commonly predicted among PASS and SEA as target molecules of lysicamine (Figure 5d), including the ATP-dependent translocase ABCB1. 

Based on the list of 36 proteins predicted by PASS (confidence value > 0.4), the top 10 significant gene ontology (GO) molecular functions presented terms related to MAP kinase, TGF-β signaling, ATP binding, and Cyclin-dependent protein kinase activity (Figure 5e). Furthermore, the significant GO biological processes predicted were related to cellular response to stimulus, signal transduction, and MAPK cascade (Figure 5f). The top ten significant GO biological processes related to the 46 proteins predicted by SEA also predicted response to stimulus and, interestingly, predicted regulation of cell death (Figure 5g). Ultimately, a protein–protein interaction network using the Search Tool for the Retrieval of Interacting Genes (STRING) database was generated for both lists (Figure 5h,i). The PASS STRING analysis shows an evident cluster of proteins associated with the MAPK pathway, and the SEA STRING is an interesting node of GAPDH, which is a protein of glycolysis metabolism.

Collectively, these results suggest that lysicamine could mechanistically modulate the cell death induced in TC cells (Figure 1 and Figure 3) via MAPK (Figure 4) or TGF-β signaling.

## 3. Discussion

Over the past decade, there has been an increase in the detected cases of TC. Some forms of TC have an unfavorable prognosis, with poor survival ranging from 0.5 to 3 years [32]. More specifically, around 50% of TC-related deaths are due to ATC and aggressive DTC cases, and despite the use of multimodal therapies, which increased survival to one year in most patients, the overall survival is still very low [33,34]. Therefore, the search for new therapeutic agents or adjuvants is imperative. Considering this, our group focused on evaluating for the first time the anti-neoplastic efficacy of lysicamine, in ATC and DTC cells in vitro. 

Lysicamine, an alkaloid oxoaporphinic, can be isolated from various herbs and plant extracts. There is limited data on the anti-cancerous properties of lysicamine in the literature. In a previous study, lysicamine showed moderate cytotoxic activity in breast and hepatocellular carcinoma cell lines as compared to doxorubicin [23]. When combined with metals, lysicamine blocked the cell cycle and induced apoptosis in liver and lung carcinoma cell lines [35]. The antioxidant activity of lysicamine without cytotoxic effect on prostate cancer and gastric adenocarcinoma cell lines was reported [24]. Conversely, lysicamine exhibited no antioxidant activity while reducing the viability of a melanoma cell line [36]. Based on these works, we hypothesized that lysicamine might exhibit antineoplastic activity in TC cells. Notably, this is the first study on lysicamine as a potential therapeutic in aggressive TC.

Our data in 2D cultures show that lysicamine decreased the viability of TC cell lines, as well as in non-tumoral thyroid cell lines (Nthy-ORI) while exhibiting anti-clonogenic properties across the TC cell lines. Our findings are consistent with results from Kang and colleagues on reduced viability of A375 melanoma cells following incubation with lysicamine [36]. Additionally, lysicamine combined with metals, such as Ru^II^, Rh^III^, Mn^II^, and Zn^II^, reduced the viability of liver, lung, and bladder cancer cell lines, as well as of non-tumor liver cancer cells [35]. These studies support the cytotoxicity potential of lysicamine in tumors of various sources. 

By calculating the SI in the 2D model, the KTC-2 cell line showed the highest selectivity of Lysicamine (SI = 1.98). No other study determined the SI of lysicamine. Due to the various accepted SI value thresholds for a bioactive and non-toxic compound (SI > 1, >2, or >10) [37], further studies are necessary to determine the lysicamine selectivity to TC cells. Our results also provide a foundation and support the possibility of designing a more selective compound based on the lysicamine formula.

Further, the effects of lysicamine on KTC-2 and HTH83 cells were evaluated in a 3D model of TC spheroids. Consistent with the literature, spheroids were more resistant to lysicamine than cells in 2D, and the viability of TC in spheroids treated with the IC_50_ concentration for 72 h was not affected. In the previous studies, prostate cancer spheroids, as well as pancreatic cancer and breast cancer spheroids, were significantly more resistant to chemotherapy when compared to cells grown as a monolayer culture [38,39,40,41]. This can be explained by the transport barriers that are present in 3D cellular structures. To overcome the transport issue, it was necessary to increase the IC_50_ concentration of lysicamine for a chronic treatment lasting 10–14 days. In this experimental setting, a significant reduction in the spheroid area in both TC spheroids was observed, confirming the cytotoxic activity of lysicamine also in 3D models. Several studies with different cell types and drugs have already demonstrated that both monolayer and 3D cultures can exhibit distinct responses, which may include increased or decreased drug sensitivity [42,43,44], as well as distinct gene and protein expression profiles, influencing the drug response [45]. These differences may depend on the feature of the 3D structure microenvironment, contributing to specific cell–cell interactions and biochemical signals, as well as to a gradient of hypoxia nutrients [46]. Spheroid structures also exhibit a distinct cell gradient, including a proliferative outer layer, a quiescent middle layer, and a necrotic core [47]. In a study investigating doxorubicin penetration in colon cancer cell spheroids (HCT 116), the doxorubicin concentration within the spheroid increased at longer drug incubation times, with significantly higher levels observed in the necrotic core after 72 h compared to the outer layers [47]. 

Our study also showed that lysicamine slightly reduced cell migration in KTC-2. No other studies have examined the impact of lysicamine on cell migration in cancer cell lines; however, research on apomorphine, another alkaloid, demonstrated its ability to suppress cell invasion, as well as TNF-α-induced MMP9 expression in breast cancer cells [48]. AKT1 has been shown to promote migration and invasion via the induction of MMP2, MMP9, and NF-κβ [49]. Knockout of AKT in colon cancer cell lines reduced cell migration rates [50]. Here, we show that AKT phosphorylation was inhibited at the IC_50_ concentration in KTC-2, suggesting that inhibition of migration mediated by lysicamine may be dependent on AKT inhibition. In HTH83, AKT phosphorylation reduction was observed at 1.5× IC_50_. Hence, it is possible that higher lysicamine concentrations may be needed to modulate migration in this ATC cell line. 

The annexin assay revealed an increase in 7AAD-stained cells, indicating that the mechanism underlying the viability reduction involves necrosis. This was further supported by no change in the apoptotic marker caspase 3/7 activity following treatment. Conversely, it was previously reported that Lysicamine-Rh^III^/Mn^II^ led to an increase in the number of apoptotic cells in liver carcinoma cell lines [35]. To the best of our knowledge, our study represents the first report demonstrating the induction of necrosis by lysicamine.

Based on the available literature suggesting lysicamine’s potential to induce apoptosis [35], we investigated whether the observed necrosis resulted from accidental cell death due to unspecific toxicity or if the treatment triggered programmed cell death via necroptosis. The mechanism of necroptosis remains incompletely understood; however, the RIPK1/RIPK3/MLK complex, known as the necrosome, plays a central role in the necroptosis signaling pathway [51]. Activated MLKL oligomerizes and translocates to the cell membrane, ultimately causing membrane rupture [51]. Therefore, the annexin assay was performed after inhibiting RIPK1 with Nec-1, which resulted in a reduction in necrotic cells, indicating that lysicamine activates the necrosome pathway, inducing necroptosis and cell death. Since resistance to apoptosis is one of the hallmarks of cancer, necroptosis is being proposed as a target to be induced in apoptotic-resistant cells, despite tumor progression activity having also been shown in some types of cancer [52]. Necroptosis is known to induce inflammation; thus, necroptosis could also be activated, taking into account that the modulation of the immune system is a recognized target for new cancer therapy [53]. In this context, a necroptosis-related gene signature in PTC patients was proposed for prognosis prediction and immunotherapy response [54]. 

The potential anti- or pro-oxidant role of lysicamine in TC cell lines was simultaneously explored, along with the involvement of ROS in cell death. Higher levels of ROS could enhance necrosome assembly and RIPK3 activity [55]. Kim and collaborators proposed a mechanism by which astaxanthin (ASX), a carotenoid, induces necroptosis in gastric cancer cell lines. High concentrations of ASX were found to increase NADPH oxidase activity, leading to elevated ROS levels and subsequently inducing the phosphorylation of RIPK1 and the formation of necrosome [53]. In our study, distinct responses in ROS generation were observed among the cell lines. Treatment with lysicamine resulted in increased ROS levels only in the HTH83 cell line (BRAF^WT^). Previous research demonstrated that the lysicamine-Rh^III^/Mn^II^ complex increased ROS levels in the liver cancer cell line (BRAF^WT^) [35]. We hypothesized that the KTC-2 and BCPAP cell lines did not exhibit an increase in ROS generation due to the presence of BRAF^V600E^ mutation. This mutation induced the expression of antioxidant genes regulated by the NRF2 pathway, as previously shown [56,57]. It has also been reported that BRAF^V600E^ inhibition reduces the expression of proteins such as GLUT1, GLUT3, and hexokinase 2. This, along with the suppression of glycolytic metabolism, leads to decreased lactate and ATP production and increased oxidative stress and ROS generation [58]. Another study proposed a negative feedback mechanism in BRAF-mutant melanoma cells to limit ROS production. This pathway involves ROS activating pyruvate dehydrogenase kinases (PDKs), which inactivate pyruvate dehydrogenase (PDH), thereby restricting pyruvate entry into the Krebs cycle and mitochondrial oxidative phosphorylation, ultimately preventing ROS from reaching toxic levels [59]. This study also suggests that combining PDKs with BRAF and MEK inhibitors can increase ROS production to toxic levels, promoting cell death [59]. 

Nevertheless, our findings revealed that lysicamine induced cell death even in the absence of increased ROS, irrespective of the BRAF mutation status. These data imply that the lysicamine mechanism of action may not directly target the overactivated BRAF in cells harboring BRAF^V600E^. To better understand the lysicamine mechanism of action, we assessed whether reducing ROS via NAC treatment could impact cell death in HTH83. Interestingly, our results demonstrated that lower ROS levels in the *BRAF*^WT^ cell line did not significantly affect cell death, confirming that ROS does not play a significant role in the necrotic process in our TC model. 

Subsequently, the potential of lysicamine to modulate the prevailing signaling pathway in TC was examined. The activation of the MAPK and PI3K-AKT pathways are the main events implicated in thyroid tumorigenesis, with BRAF and RAS mutations being the most prevalent drivers [60]. During TC progression and due to cancer genomic instability across more dedifferentiated TC histologies, the prevalence of mutations in genes linked to SWI/SNF chromatin remodeling complex and the PI3K/AKT/mTOR, MAPK, and JAK/STAT pathways increased [61]. Mutations in TERT, AKT1, PIK3CA, and EIF1AX were frequently co-mutated with BRAF^V600E^ and *RAS* mutations in advanced DTCs and ATC [62]. Remarkably, our findings show lysicamine ability to diminish the activation of the PI3K/AKT pathway by suppressing AKT phosphorylation in anaplastic cell lines characterized by BRAF and RAS driver mutations. Inhibition of PI3K and AKT reduces proliferation and motility in different human TC cells. AKT inhibitors induce apoptosis and inhibit de progression of the cell cycle progression at G2/M [63].

Preclinical trials have been revealing beneficial results due to MAPK inhibitors associated with PI3K pathway inhibitors, TKIs, or even NF-kβ inhibitors [64]. The combination of dactolisib (a PI3K inhibitor) and RAF265 (an RAF inhibitor) caused a pronounced inhibition of proliferation in DTC cell lines harboring RAS, BRAF, RET, and PTEN mutations [65]. Similar results were observed in ATC and FTC cell lines treated with sorafenib combined with dactosilib, which enhanced apoptosis when compared to sorafenib alone [66]. Still, the use of palbociclib (a CDK4/6 inhibitor) associated with omipalisib (a PI3K/mTOR dual inhibitor) in ATC in vitro and in vivo models significantly reduced cell proliferation in cell lines and inhibited tumor growth in lower doses [67]. Conversely, in our study, lysicamine exhibited no impact on reducing MAPK activation in TC cells. Importantly, we investigated the classical MAPK pathway, verifying the phosphorylation status of ERK1/2 after lysicamine treatment, while our in silico analysis suggested the involvement of proteins belonging to other MAPK families, such as the JNK and p38/SAPK pathways [68]. This contrast in data highlights the complexity of lysicamine’s effects on various components of the MAPK pathway, which warrants further investigation.

Ultimately, an in silico analysis was performed to identify potential target proteins and interactions of lysicamine. Among the proteins identified, several members of the MAPK family, including MAPK3 (ERK1), were listed; however, in our ATC model, no ERK1 modulation by lysicamine treatment was confirmed. This analysis also suggested that the TGF-β signaling pathway is worthy of further exploration. This pathway is known to be involved in multiple processes, including tumor growth suppression, thyroid cell dedifferentiation, epithelial-mesenchymal transition, and cancer progression. For example, crosstalk between the TGF-β and AKT/mTOR pathways has been shown to regulate many cellular functions [69]. Furthermore, it was shown that the inhibition of both MAPK and pSMAD activation induced by TGF-β/activin leads to enhanced redifferentiation and increased radioiodine uptake in cancer cells, potentially having a therapeutic effect [70]. Additionally, the in silico analysis identified RIPK4, a protein from the same family as RIPK1 and RIPK3, known for its role as an activator of NF-κβ signaling [71]. Silencing of RIPK4 in melanoma cell lines had no impact on apoptosis or necroptosis induction but reduced phosphorylation of Akt [72]. In ovarian cancer, elevated RIPK4 expression has been correlated with more adverse progression, and its inhibition decreased metastasis in in vitro experiments [73]. Moreover, our analysis revealed that lysicamine has the potential to interact with GAPDH, a protein associated with glycolysis metabolism. This finding may indicate how lysicamine could induce cell death via metabolic changes [74]. 

In conclusion, our study has shown, for the first time, the potential antineoplastic activity of lysicamine via the inhibition of the AKT pathway and the induction of necrosis/necroptosis in ATC cell lines. These finding may have significant implications, particularly regarding the prospect of synergistic therapies involving drugs aimed at modulating diverse pathways, and highlights lysicamine as a promising compound for TC treatment, especially in advanced stages, which should be explored in future studies.

## 4. Materials and Methods

### 4.1. Cells, Reagents, and Treatments 

Lysicamine was synthesized via benzyne chemistry, as previously described [27].

Two ATC cell lines (KTC-2 and HTH83) and one PTC cell line (BCPAP) donated by Dr. Edna Kimura were cultured in RPMI supplemented with 5% of Fetal Bovine Serum (FBS) (ThermoFisher Scientific, Waltham, MA, USA) for KTC-2, and with 10% of FBS for HTH83 and BCPAP; 0.01 µg/mL of penicillin–streptomycin (ThermoFisher Scientific, Waltham, MA, USA) at 37 °C; and 5% of CO_2_ [75]. The medium was changed every 48 h, and when necessary, cells were detached with Trypsin-EDTA (0.25%) (ThermoFisher Scientific, Waltham, MA, USA).

To assemble the 3D TC spheroids, the hanging drop technique with methylcellulose (M7027, Sigma-Aldrich St. Louis, MO, USA) was employed as we previously described [41]. Briefly, drops of 20 µL containing 3000 cells in RMPI supplemented with FBS and 0.24% methylcellulose were placed on the cover of the culture plate and incubated inverted for 96 h at 37 °C and 5% CO_2_. Next, spheroids were transferred to a 96-well low-attachment plate (Greiner Bio-One, Kremsmünster, Austria). Spheroid morphology was evaluated visually using light micrographs taken at 10× magnification. Microscope photos were used to determine areas via ImageJ software (Version 1.53t, imagej.nih.gov) analyses. 

Lysicamine was resuspended in DMSO (Merck KGaA, Darmstadt, Germany) to a final concentration of 3000 µg/mL. As an exploratory experiment, cells were treated for 48 h with 15 µg/mL (51.49 µM), 30 µg/mL (102.99 µM), and 60 µg/mL (205.97 µM) of lysicamine or DMSO as vehicle. After IC_50_ determination, treatments were performed using IC_50_, 0.5× IC_50_, 1.5× IC_50_, and 2× IC_50_ concentrations. 

### 4.2. Cell Viability, Proliferation, and IC_50_ and SI Determination 

Cell viability was determined using PrestoBlue Cell Viability Reagent (ThermoFisher Scientific, Waltham, MA, USA), following the manufacturer’s instructions. Briefly, cell lines were seeded into 96-well plates at an initial density of 3000 cells and incubated at 37 °C, 5% CO_2_ for 24 h. After 72 h of treatment, 10% of PrestoBlue reagent was added to each well and incubated for 90 min at 37 °C and 5% CO_2_, protected from light. Then, fluorescence (540 nm excitation/590 nm emission) was measured using a microplate reader M3 and SoftMax Pro 7 Software (Molecular Devices, San José, CA, USA). The relative fluorescence of treated cells was normalized against the value of the vehicle-treated control cells (DMSO) and expressed as a percentage of viable cells. Four biological replicates were performed in triplicate.

The 50% inhibitory concentration (IC_50_) required to achieve 50% cell death was obtained from non-linear regression analysis of the dose–response curve using GraphPad Prism 5 (version 5.01). Briefly, cells were treated for 72 h with a serial dilution of lysicamine starting from 51.5 µM. The SI value was determined as the ratio of the non-tumoral NthyORI IC_50_ against the cancer cell IC_50_ values. Additionally, the IC_50_ of cisplatin was calculated as a positive control. Three to five biological replicates were performed in triplicate.

For the experiments in 3D TC spheroids, each spheroid was subjected to a treatment with IC_50_ or twice IC_50_ (2× IC_50_) concentration of lysicamine in the 96-well low attachment plate. Fresh media with lysicamine was added at days 3, 7, and 10, and the viability was evaluated at days 3, 5, 7, 10, and 14. After each treatment, 10% of PrestoBlue reagent was added to each well and incubated for 5 h at 37 °C and 5% CO_2_, protected from light. Fluorescence (540 nm excitation/590 nm emission) was measured as described earlier. Four biological replicates were performed in triplicate.

### 4.3. Clonogenic Assay

To evaluate colony formation after treatment, 250 cells were seeded on a 6-well plate and incubated for 24 h at 37 °C, 5%CO_2_. Next, the cells were treated with lysicamine IC_50_ for 72 h and then cultured for 10–15 days, changing media, without treatment, twice a week. Next, each well was washed with 1 mL of PBS, fixed with methanol (33%) in an acetic acid (33%) water solution for 10 min at 4 °C, and then stained with 0.5 mL of crystal violet (1%) for another 10 min at room temperature. The excess of crystal violet was washed with distilled water, and plates were allowed to dry at room temperature. The number of colonies was counted manually and compared with the number of DMSO treatment cell colonies. Two biological replicates were performed in duplicate. 

### 4.4. Cell Death Assays

Annexin V-FITC/7-Aminoactinomycind D (7-AAD) kit (ThermoFisher Scientific, Waltham, MA, USA) was used to assess whether lysicamine induces apoptosis. Briefly, 2 × 10^5^ cells were seeded into a 6-well plate and incubated for 24 h. Cells were treated with lysicamine at IC_50_ for 48 h, with or without N-acetylcysteine (NAC, 5 mM) (Sigma-Aldrich St. Louis, MO, USA) to remove reactive species of oxygen (ROS). Cells were detached with trypsin-EDTA (0.25%), counted, and 1 × 10^5^ cells were incubated for 30 min, in the dark, at 37 °C with Annexin V-FITC (0.2 mg/mL) and 7-AAD. Then, samples were analyzed via flow cytometry (488 nm) (BD AccuriTM C6 Flow Cytometer, Becton Dickinson, Franklin Lakes, NJ, USA). Three to eight biological replicates were performed.

To evaluate caspase activity, CellEvent™ Caspase-3/7 Green Detection Reagent (ThermoFisher Scientific, Waltham, MA, USA) was used. Cells grown in a 24-well plate (1 × 10^5^/well) for 24 h were treated with lysicamine at IC_50_ for 48 h. Cells were detached and incubated for 30 min, in the dark, at 37 °C with 2 mM of CellEvent^TM^. Then, samples were analyzed in a flow cytometer (488 nm) (BD AccuriTM C6 Flow Cytometer). Results were determined using FlowJo software (version 10.8.1). Four to eight biological replicates were performed.

To assess the modulation of necroptosis, 2 × 10^5^ cells plated in 6-well plates for 24 h were treated with lysicamine at IC_50_ in the presence or absence of 30 µM of necrostatin-1 (Nec-1) (480066, Sigma-Aldrich St. Louis, MO, USA), a necrosome inhibitor. After 48 h, the cells were subjected to the annexin assay. Three biological replicates were performed.

### 4.5. Assessment of Reactive Oxygen Species (ROS)

To quantify the production of ROS after treatment, 2 × 10^5^ cells were seeded into a 6-well plate and incubated for 24 h at 37 °C, 5%CO_2_. Cells were treated with lysicamine at IC_50_, with or without NAC (5 mM) to remove ROS, and incubated for 48 h at 37 °C, 5% CO_2_. Then, the media was removed, and cells were washed with PBS before the addition of 2′,7′-dichlorofluorescin diacetate (DCF-DA) to a final concentration of 25 µM (Sigma-Aldrich St. Louis, MO, USA). After 30 min of incubation at 37 °C, cells were washed with PBS, detached from the plate, centrifuged at 500× *g* for 5 min, and resuspended in PBS. ROS production was evaluated via flow cytometry (488 nm) (BD AccuriTM C6 Flow Cytometer), and mean fluorescence intensity values were expressed as percentages. Four to eight biological replicates were performed.

### 4.6. Computational Target Prediction

The prediction of the mechanism regarding the biological activity of lysicamine was performed as previously described [76]. Briefly, two in silico tools, PASS (http://way2drug.com/PassOnline, accessed on 30 March 2022) and SEA (https://sea.bkslab.org, accessed on 30 March 2022), were used. A confidence value of less than 0.4 was considered for proteins with a direct interaction predicted by PASS. No cutoff value was applied for proteins predicted by SEA, considering the restricted number of predicted proteins. A protein–protein interaction network for the specific proteins was constructed using the database Search Tool for the Retrieval of Interacting Genes (STRING—version 10.5; http://string-db.org/, accessed on 30 March 2022). Pathway enrichment analyses were performed using the GO-Biological Process.

### 4.7. AKT and ERK Pathways Analysis

To verify the modulation of the MAPK and PI3K/AKT pathways by lysicamine, a Western blotting assay was performed. For this purpose, 2 × 10^5^ cells plated in 6-well plates for 24 h were treated with 0.5× IC_50_, IC_50_, or 1.5× IC_50_ lysicamine or control in serum-starved media overnight. Next, the medium was removed, and a serum stimulus (10% FBS) with 0.5× IC_50_, IC_50_, or 1.5× IC_50_ lysicamine concentration or control was added, and cells were incubated for 30 min.

The cells were harvested with ice-cold lysis buffer (10 mM NaCl, 50 mM Tris-HCl, 0.5% NP40, 0.1 mM EDTA, 0.1 M NaVO_4_, complete protease inhibitor, Mini, Roche, 1×). After centrifugation (10,000 rpm, 5–10 min, 4 °C), the protein concentration was measured using the Qubit™ Protein and Protein Broad Range (BR) Assay Kit (ThermoFisher Scientific, Waltham, MA, USA). The total protein extract (50 μg) was separated using a 10% polyacrylamide gel and transferred to a PVDF membrane. Membranes were blocked in TBS 5% skimmed milk solution for 30 min at room temperature. Furthermore, they were incubated for 16 h at 4 °C with appropriated primary antibodies, as described in the following: Phospho-p44/42 MAPK (Erk1/2) (Thr202/Tyr204) Rabbit (P-MAPK) (1:3000 dilution) and Phospho-Akt (Ser473) (D9E) XP^®^ Rabbit (dilution 1:1000) (Cell Signaling Technology, Danvers, MA, USA). All were diluted in TBS with 5% BSA. Next, membranes were incubated for 40 min with the following secondary antibodies: anti-rabbit secondary antibody conjugated to horseradish peroxidase (HRP) (dilution 1:5000) or anti-mouse conjugated to HRP (1:10,000) (Sigma-Aldrich St. Louis, MO, USA). Proteins were detected using SuperSignal™ West Atto Ultimate Sensitivity Substrate (ThermoFisher Scientific, Waltham, MA, USA) and visualized with the iBright CL750 Imaging System photodocumenter (ThermoFisher Scientific, Waltham, MA, USA). 

### 4.8. Cell Migration

To determine if lysicamine affects cell migration, 2 × 10^5^ cells were seeded into a 24-well plate and incubated for 48 h at 37 °C, 5% CO_2_. First, the cells were treated with mitomycin C for 2 h to inhibit cell proliferation. Then, a line was scratched with a pipette tip, and the cells were washed with PBS, and fresh media with IC_50_ lysicamine were added. The plate was incubated at 37 °C, 5%CO_2_. Images were taken by Zeiss Axio microscope every 6 h until 48 h from the time of treatment. Using ImageJ software, the difference between the areas over time was calculated as a percentage relative to the initial wound area. Two biological replicates were performed in duplicate.

### 4.9. Statistical Analysis

All statistical analysis was performed in GraphPad Prism 5. Significance was determined using *t*-Test, two-way or one-way ANOVA with Bonferroni or Turkey post-tests; graphs were plotted as mean + SEM.

## Figures and Tables

**Figure 1 pharmaceuticals-16-01687-f001:**
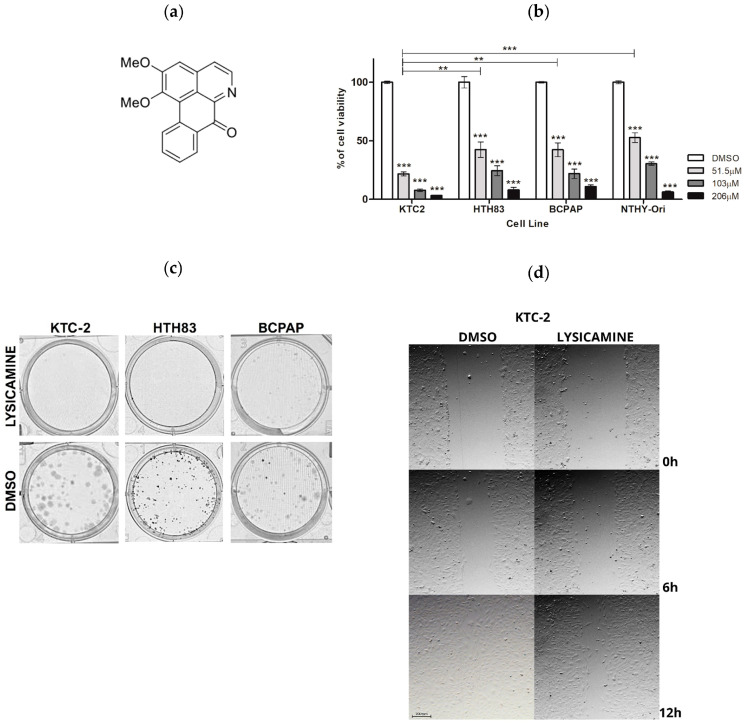
Lysicamine induces cell toxicity and reduces migration in thyroid tumor cells. (**a**) Molecular structure of lysicamine, synthesized via benzyne chemistry [27]. (**b**) A reduction in cell viability was observed in KTC-2, HTH83, BCPAP (TC cell lines), and Nthy-ORI (thyroid non-tumoral cell line) 72 h after treatment with three concentrations of lysicamine. Cell viability was measured via PrestoBlue assay and expressed as a percentage of viability compared to non-treated cells. (**c**) Representative images of colony assay across all TC cell lines 72 h after lysicamine IC_50_ treatment vs. no treatment control (DMSO) and subsequent 10 days of culture without treatment. (**d**) Representative images of wound closure progression 12 h after lysicamine treatment of KTC-2 at IC_50_ concentration. For (**b**), data were plotted as mean ± SEM of at least three biological replicates in triplicate; significance was determined by two-way ANOVA multiple comparisons followed by Bonferroni test (post-test); ** *p* < 0.001 *** *p* < 0.0001.

**Figure 2 pharmaceuticals-16-01687-f002:**
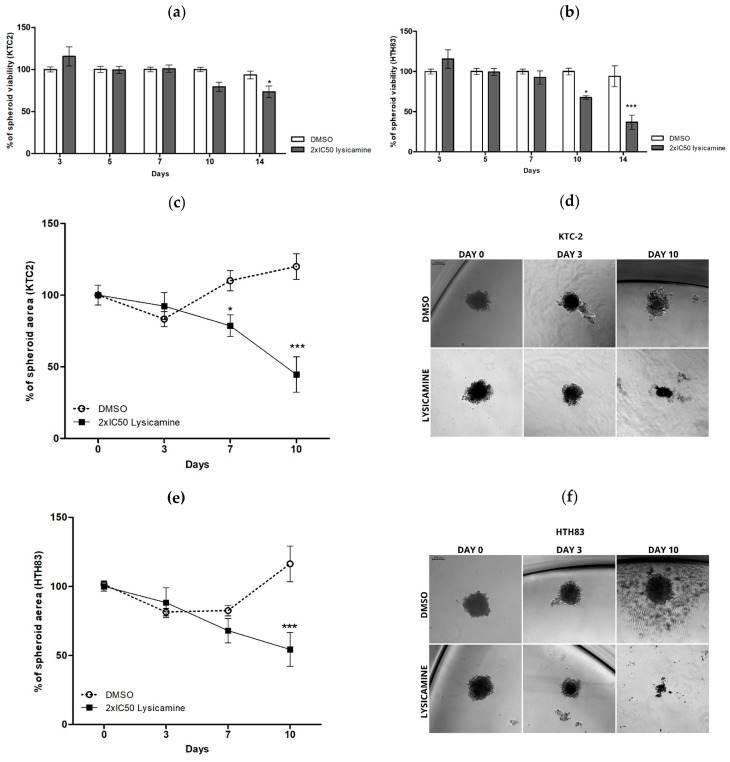
Lysicamine induces cell toxicity and reduces thyroid cancer cell spheroids area. (**a**,**b**) A reduction in cell viability was observed in spheroids of KTC-2 (**a**) on day 14 and in HTH83 (**b**) on days 10 and 14 after 2× IC_50_ lysicamine treatment. Spheroid viability was measured via PrestoBlue assay and expressed as a percentage compared to untreated control. (**c**,**e**) Reduction in KTC-2 (**c**) and HTH83 (**e**) spheroids area after 2× IC_50_ lysicamine treatment. (**d**,**f**) Representative images of KTC2 (**d**) and HTH83 (**f**) spheroids at days 0, 3, and 10 after treatment with 2× IC_50_ lysicamine vs. DMSO. The area of the spheroids from the images was quantified by ImageJ, Version 1.53t (imagej.nih.gov) and expressed as a percentage of untreated control. Data were plotted as mean ± SEM of at least three biological replicates in triplicate; significance was determined by two-way ANOVA multiple comparisons followed by Bonferroni test (post-test); * *p* < 0.05; *** *p* < 0.0001.

**Figure 3 pharmaceuticals-16-01687-f003:**
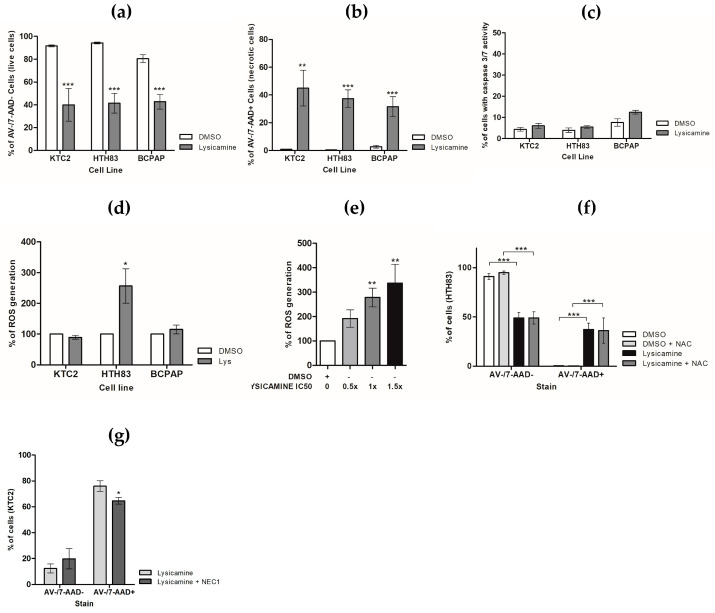
Lysicamine induces necroptosis in TC cells in a ROS-independent Manner. (**a**) Live cells (AV−/7-AAD−) were decreased in all TC cell lines by Lysicamine; (**b**) the same treatment resulted in increased necrotic cells (AV−/7-AAD+) in all TC cell lines; and (**c**) had no effect on Caspase 3/7 activation in all TC cell lines. (**d**) Lysicamine increased ROS production only in HTH83 (**e**) in a concentration-dependent manner. (**f**) In HTH83, ROS inhibition by NAC did not increase live cells (AV−/7-AAD−) or reduce necrotic cells (AV−/7-AAD+). (**g**) Necrosome inhibition via NEC1 treatment reduced necrotic cells (AV−/7-AAD+) triggered by lysicamine treatment. In all experiments, cells were treated with lysicamine IC_50_ or DMSO (control) for 48 h, stained, and analyzed via flow cytometry. Results were expressed as percentages compared to untreated controls. Data were plotted as mean ± SEM of at least three biological replicates in triplicate; significance was determined by one-way or two-way ANOVA multiple comparisons followed by Bonferroni test (post-test); * *p* < 0.05, ** *p* < 0.001, and *** *p* < 0.0001.

**Figure 4 pharmaceuticals-16-01687-f004:**
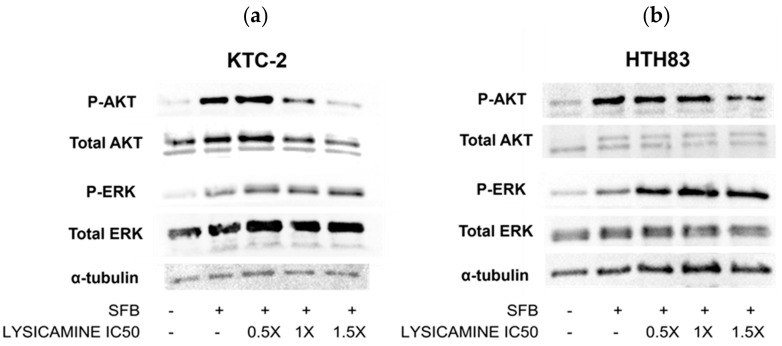
Lysicamine modulates AKT phosphorylation in thyroid cancer cells. (**a**) p-AKT was reduced in KTC-2 in a dose-dependent manner after lysicamine treatment. (**b**) p-AKT was reduced in HTH83 cells with 1.5× IC_50_ lysicamine treatment. No ERK phosphorylation modulation was observed in both cell lines. Cells were cultured without fetal bovine serum (FBS), and after 24 h, FBS was added alone or combined with DMSO or lysicamine 0.5×, 1×, or 1.5× IC_50_0. Protein extraction was performed 30 min treatment after. Representative images of two or more biological replicates conducted per cell line are presented.

**Figure 5 pharmaceuticals-16-01687-f005:**
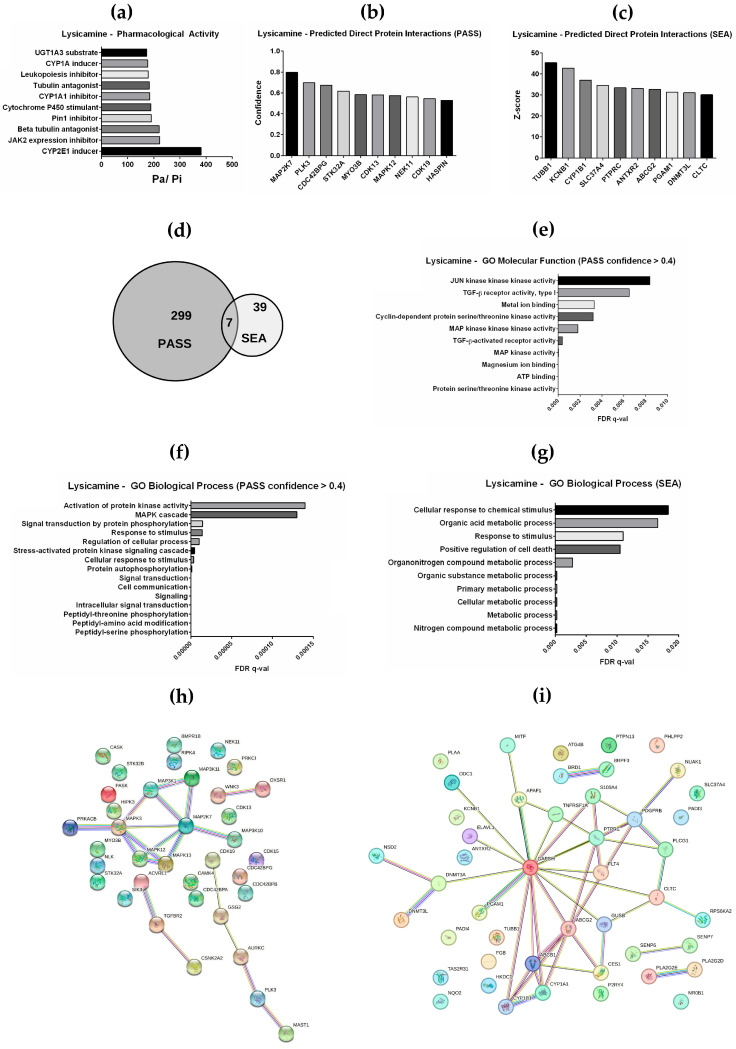
Lysicamine activity was characterized via in silico analysis. (**a**) PASS analysis resulted in a list of pharmacological activities. (**b**) PASS analysis also predicted 36 protein interactions (confidence value > 0.4). (**c**) SEA analysis predicted 46 direct protein interactions. (**d**) Among both lists, 7 proteins were shared. Go Biological Process analysis modeled (**e**) molecular functions and (**f**) biological processes for proteins predicted by PASS and (**g**) biological processes by SEA. STRING database was used to generate a protein–protein interaction network for proteins predicted (**h**) by PASS and (**i**) by SEA.

**Table 1 pharmaceuticals-16-01687-t001:** Lysicamine IC_50_ values and a corresponding selectivity index (SI) calculated from dose–response curve 72 h after treatment.

Cell Line	Lysicamine IC_50_ (µM) ± SD	Selectivity Index (SI)
KTC-2 (ATC)	15.6 ± 2.1	1.98
HTH83 (ATC)	36.4 ± 11.6	0.85
BCPAP (PTC)	30.5 ± 13.2	1.01
NThy-ORI (non-tumoral)	30.9 ± 4.8	-

## Data Availability

Data are contained within the article and Appendix A.

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
