# Peer review of "Lysicamine Reduces Protein Kinase B (AKT) Activation and Promotes Necrosis in Anaplastic Thyroid Cancer"

_pharmaceuticals, 2023, doi:10.3390/ph16121687_

Round 1

Reviewer 1 Report

Comments and Suggestions for Authors

This is a piece of new and incremental addition on the data, effect of Lysicamine. 

1. The authors focus anapestic thyroid carcinoma and mechanistic basis in vitro. That is fine, but I wonder, as the authors state in the introduction the effect of Lysicamine is different from Breast cancer, liver cancer (effective) vs gastric cancer (non-effective). At first the authors make sure the assay system is comparable with the previous data. I suggest the authors must check the cell lines of breast cancer or liver cancer and/or gastric cancer in their own system and replicate the variation of the effectiveness of Lysicamine in these cell lines.

2. I agree with that ATC and DTC are different in pathology, clinical, prognosis, and response to therapy, but Lysicamine effect may not be discussed only in the lines of thyroid cancer biology.

3. The chemicals from natural product is often not familiar to the readers outside the field. The dosages used in these in vitro experiment and in human use as a traditional medicine are comparable?

Author Response

First and foremost, we appreciate all the comments and the time the reviewer dedicated to reviewing our manuscript.

  1. The authors focus anapestic thyroid carcinoma and mechanistic basis in vitro. That is fine, but I wonder, as the authors state in the introduction the effect of Lysicamine is different from Breast cancer, liver cancer (effective) vs gastric cancer (non-effective). At first the authors make sure the assay system is comparable with the previous data. I suggest the authors must check the cell lines of breast cancer or liver cancer and/or gastric cancer in their own system and replicate the variation of the effectiveness of Lysicamine in these cell lines.

The aim of this study was to contribute new knowledge about lysicamine's activity in thyroid cancer cells. We were not interested in comparing our model with those previously described; thus, the sentence that could lead to this observation was removed. Instead, more information about lysicamine activity was added to the introduction section improving this section.

  1. I agree with that ATC and DTC are different in pathology, clinical, prognosis, and response to therapy, but Lysicamine effect may not be discussed only in the lines of thyroid cancer biology.

We compared our data with those previously reported for other cancer cell types in the discussion section. It's important to note that, up to now, few studies have been published demonstrating its cytotoxic effect in cancer cells. We have added more information in the introduction section, and from line 431, we had previously discussed the available data so far.

  1. The chemicals from natural product is often not familiar to the readers outside the field. The dosages used in these in vitro experiments and in human use as a traditional medicine are comparable?

It is very difficult to compare the action of an isolated compound, as in our study, with the action of an extract or tea from different species or with a mix of herbs, as used in Traditional Chinese Medicine.

Reviewer 2 Report

Comments and Suggestions for Authors

In the present manuscript, the authors investigated the effects of lysicamine on thyroid cancer cell lines. The authors have performed extensive in vitro and in silico studies demonstrating its potential as a promising compound for thyroid cancer treatment. The manuscript is well written, and the experiments appear to have been properly conducted, leading to valid results and properly interpreted conclusions.

I would like to point out a few points that I think will benefit the manuscript:

The number of replicates should be given for each in vitro experiment.

Regarding the effect of lysicamine on the area of tumor spheroids, the authors took photos on different days, with day 10 being the last day. Did you perhaps take the photos on day 14? It would be interesting to see what they look like on day 14, since the viability of the spheroids was affected by lysicamine on day 14.

Why didn't the authors perform all the experiments on the BCPAP cell line as well?

In Section 2.5, the authors predicted the pharmacological activity and interacting proteins of lysicamine. They concluded that lysicamine could mechanistically modulate cell death induced in TC cells through MAPK or TGF-β signaling. However, in Section 2.4, the results show that lysicamine does not modulate MAPK signaling. Please explain these contradictory results.

Author Response

First and foremost, we appreciate all the comments and the time the reviewer dedicated to reviewing our manuscript.

  • The number of replicates should be given for each in vitro experiment.

We added the number of replicates on the methods section.

  • Regarding the effect of lysicamine on the area of tumor spheroids, the authors took photos on different days, with day 10 being the last day. Did you perhaps take the photos on day 14? It would be interesting to see what they look like on day 14, since the viability of the spheroids was affected by lysicamine on day 14.

We agree with the reviewer's suggestion that the day 14 photo would have provided a more comprehensive view of the experiment. However, regrettably, we lost the spheroid plates on day 14. Nevertheless, the photos taken on day 10 clearly illustrate Lysicamine's activity on the 3D structures. Additionally, for HTH83, the viability reduction effect was also significant on day 10.

  • Why didn't the authors perform all the experiments on the BCPAP cell line as well?

We initiated the study with PTC and ATC cells. However, as the study progressed, we concentrated our efforts on investigating the mechanisms in the most aggressive forms of thyroid cancer, specifically ATC. By doing so, we aimed to gather more comprehensive data about lysicamine and its effects on thyroid cancer, which we can now share with the scientific community.

  • In Section 2.5, the authors predicted the pharmacological activity and interacting proteins of lysicamine. They concluded that lysicamine could mechanistically modulate cell death induced in TC cells through MAPK or TGF-β signaling. However, in Section 2.4, the results show that lysicamine does not modulate MAPK signaling. Please explain these contradictory results.

We agree with the reviewer that this data seems contradictory. It is important to note that our investigation primarily focused on the classical MAPK pathway, particularly through the activation of ERK1/2, while the in silico analysis showed activation of proteins belonging to other MAPK families, such as JNKs and p38/SAPKs pathways. (https://www.kegg.jp/pathway/hsa04010, doi: 10.1101/cshperspect.a011254).

Additionally, this information was included into the discussion section.

Reviewer 3 Report

Comments and Suggestions for Authors

The manuscript reported lysicamine as a potential antineoplastic compound in ATC cells by inhibiting AKT activation and inducing cell death. In general, this study was not rigorously designed. I think the manuscript can be suitable or publication in Pharmaceuticals after major revision.

1. Why some positive control is not used during cytotoxic and other assays? Results of positive control should be included in figure 1.

2. In the cytotoxicity test, the IC50 of lysicamine against KTC-2 cells was 15.6 µM, indicating that the activity of lysicamine on anaplastic thyroid cancer was actually not good.

3. Wound healing experiments showed that lysicamine had no obvious effect on the migration ability of KTC-2 cells.

4. The results of prediction of pharmacological activity and interacting proteins suggest that lysicamine could mechanistically modulate the cell death induced in TC cells through MAPK or TGF-β signaling. However, the authors did not verify the effects of the compounds on MAPK and TGF-β protein expression through Western blotting experiments.

5. Some of the figure are not nicely layout and has low resolution.

6. Throughout the manuscript, typing and grammar errors must be checked. For example:

In 87 line, “50” of “IC50” should be subscripted.

In 88 line, “15.6µM should be changed to “15.6 µM

These are just some examples.

Comments on the Quality of English Language

No

Author Response

First and foremost, we appreciate all the comments and the time the reviewer dedicated to reviewing our manuscript. We inform that Introduction was improved.

  1. Why some positive control is not used during cytotoxic and other assays? Results of positive control should be included in figure 1.

We added the positive control cisplatin IC50 values in the Results section.

  1. In the cytotoxicity test, the IC50 of lysicamine against KTC-2 cells was 15.6 µM, indicating that the activity of lysicamine on anaplastic thyroid cancer was actually not good.

 We agree with the reviewer, the IC50 of lysicamine are not ideal and are higher than the values of Cisplatin. However, it is essential to consider that this study represents the first exploration of lysicamine's antitumoral activity and its mechanism of action in thyroid cancer cells. Therefore, these original findings can serve as a foundation for future research projects. For instance, there is potential to enhance lysicamine's cytotoxicity through molecule modifications or by combining it with other inhibitors, such as a BRAF inhibitor, to seek synergistic effects. Additionally, a previous study reported an improvement in the cytotoxic effect of lysicamine when combined with metal complexes in gastric cell lines (HepG2) (10.18632/oncotarget.19584), suggesting that high IC50 values should not be dismissed for future studies.

  1. Wound healing experiments showed that lysicamine had no obvious effect on the migration ability of KTC-2 cells.

Indeed, the effect of lysicamine on KTC2 was subtle but became significant at 12 hours. We decided to include this result in the manuscript. There was a sentence on line 476 that was fixed to assure that lysicamine have a slightly effect on migration.

  1. The results of prediction of pharmacological activity and interacting proteins suggest that lysicamine could mechanistically modulate the cell death induced in TC cells through MAPK or TGF-β signaling. However, the authors did not verify the effects of the compounds on MAPK and TGF-β protein expression through Western blotting experiments.

We investigated the potential of lysicamine to modulate the prevailing signaling pathway in TC. First we explored the classical MAPK pathway through the activation of ERK1/2 using Western Blot analysis. Importantly, our in silico analysis showed activation of proteins belonging to other MAPK families, the JNKs and p38/SAPKs pathways (https://www.kegg.jp/pathway/hsa04010, doi: 10.1101/cshperspect.a011254). The study of the effect of Lysicamine on other MAPK family pathways and the TGF-beta pathway will be pursued in a new project.

We also investigated the modulation of the PI3K/AKT pathway by western blotting, which is a significant finding, especially considering that this pathway is activated during the progression of thyroid cancer (doi 10.1530/JOE-17-02661).

  1. Some of the figure are not nicely layout and has low resolution.

The layout of the figures, and STRING figure resolution were improved.

  1. Throughout the manuscript, typing and grammar errors must be checked. For example:

In 87 line, “50” of “IC50” should be subscripted.

In 88 line, “15.6µM” should be changed to “15.6 µM”

These are just some examples.

A grammar and typing review was conducted by a native English speaker, and the changes were highlighted in the text.

Reviewer 4 Report

Comments and Suggestions for Authors

Dear Editor. The authors submitted the MS “Lysicamine Reduces AKT Activation and Promotes Necrosis in Anaplastic Thyroid Cancer” Marina Teixeira Rodrigues, Ana P. P. Michelli, Gustavo Felisola Caso, Dorival Mendes Rodrigues-Junior, Mirian Galliote Morale, Joel Machado-Júnior, Karina Bortoluci, Rodrigo E. Tamura, Tamiris R. C. Silva8, Cristiano Raminelli, Eric Chau, Biana Godin, Jamile Calil-Silveira, Ileana G. S. Rubio. By possible publication at the Journal. Pharmaceuticals 2023, 16, x. https://doi.org/10.3390/xxxxx.  The aim of the authors is to explore the antineoplastic effects of lysicamine on papillary TC (BCPAP) and ATC (HTH83 and KTC-2) cells. Their studies demonstrated that lysicamine act as a potential antineoplastic compound in ATC cells by inhibiting AKT activation and inducing cell death. Lysicamine treatment reduced cell viability, motility, colony formation, and AKT activation while increasing the percentage of necrotic cells. The absence of caspase activity confirmed apoptosis-independent cell death. Necrostatin-1 (NEC-1)-mediated necrosome inhibition lessened lysicamine-induced necrosis in KTC-2, hinting at potential necroptosis induction through a reactive oxygen species (ROS)-independent mechanism. Silico analysis predicted lysicamine target proteins that related to MAPK and TGF-β signaling.

The MS is well written, ease to understand, clearly presented. Abstract is well written, Introduction is concise and end with their aim. Results well described and well interpreted. Material and Methods and References well done, not authors self-references. Written a very good conclusion. Title is OK for the content of the results described. I suggest publishing at the Journal will be good for the Journal and audience.

Author Response

We truly appreciate all the comments and the time the reviewer dedicated to reviewing our manuscript.

Reviewer 5 Report

Comments and Suggestions for Authors

Anaplastic thyroid cancer (ATC) is an aggressive form of thyroid cancer (TC), accounting for 50% of total TC-related deaths. The study aims to explore the antineoplastic effects of lysicamine on papillary TC (BCPAP) and ATC (HTH83 and KTC-2) cells. Lysicamine treatment showed various effects including reduced cell viability, motility, colony formation, and AKT activation. It increased the percentage of necrotic cells without caspase activity, hinting at apoptosis-independent cell death. Furthermore, the paper suggests potential necroptosis induction via a ROS-independent mechanism. Overall, the paper is well-organized and designed. Comments and suggestions as shown below:

1. Consider clarifying the potential clinical relevance of lysicamine. How far is it from being a therapeutic option?

2. More context on the AKT pathway's significance in thyroid cancers might be helpful.

3. While the necroptosis findings are intriguing, any potential downstream effects or how this could be leveraged for treatments would add value.

4. Address any adverse effects or toxicity of lysicamine on normal cells or its potential impact on the patient's overall health.

5. While lysicamine's use in Traditional Chinese Medicine (TCM) is mentioned, the context or specific uses aren't detailed. A brief insight could be valuable.

6. There's mention of varying effects of lysicamine in 2D vs. 3D models, and in different cell lines. Some clarity on why these differences occur would be beneficial.

Author Response

First and foremost, we appreciate all the comments and the time the reviewer dedicated to reviewing our manuscript.

  1. Consider clarifying the potential clinical relevance of lysicamine. How far is it from being a therapeutic option?

There is a long way to go for Lysicamine being a therapeutic option. Up to now, few studies have been published showing its cytotoxic effect in cancer cells. The toxicity, safety and other Lysicamine effects must be investigated before being considered as an antitumoral drug. However, this study shows its potential, and serve as a basis for new projects with this purpose.

  1. More context on the AKT pathway's significance in thyroid cancers might be helpful.

More context was given in the discussion section.

  1. While the necroptosis findings are intriguing, any potential downstream effects or how this could be leveraged for treatments would add value.

More information was given in the discussion section.

  1. Address any adverse effects or toxicity of lysicamine on normal cells or its potential impact on the patient's overall health.

 Lysicamine is known to occur in low concentrations in plants, and specific data about its effects on patients are difficult to evaluate. This challenge arises because people often consume teas or extracts that may contain lysicamine. However, there is a research that evaluated the effects of lysicamine in combination with metal complexes in xenografts, and it demonstrated positive outcomes, including tumor growth reduction and a better safety profile than cisplatin (10.18632/oncotarget.19584). No other study evaluated Lysicamine adverse effects. Hence new studies must be proposed considering its potential adverse effects and clinical use.

  1. While lysicamine's use in Traditional Chinese Medicine (TCM) is mentioned, the context or specific uses aren't detailed. A brief insight could be valuable.

We have removed this phrase from the discussion section and included it in the introduction into a new context.

  1. There's mention of varying effects of lysicamine in 2D vs. 3D models, and in different cell lines. Some clarity on why these differences occur would be beneficial.

We include new information in the discussion section, line 468